# Learning Priors for Adversarial Autoencoders

## Abstract

Most deep latent factor models choose simple priors for simplicity, tractability or not knowing what prior to use. Recent studies show that the choice of the prior may have a profound effect on the expressiveness of the model, especially when its generative network has limited capacity. In this paper, we propose to learn a proper prior from data for adversarial autoencoders (AAEs). We introduce the notion of code generators to transform manually selected simple priors into ones that can better characterize the data distribution. Experimental results show that the proposed model can generate better image quality and learn better disentangled representations than AAEs in both supervised and unsupervised settings. Lastly, we present its ability to do cross-domain translation in a text-to-image synthesis task.

## 1 Introduction

Deep latent factor models, such as variational autoencoders (VAEs) and adversarial autoencoders (AAEs), are becoming increasingly popular in various tasks, such as image generation (Larsen et al., 2015), unsupervised clustering (Dilokthanakul et al., 2016; Makhzani et al., 2015), and cross-domain translation (Wu et al., 2016). These models involve specifying a prior distribution over latent variables and defining a deep generative network (i.e., the decoder) that maps latent variables to data space in stochastic or deterministic fashion. Training such deep models usually requires learning a recognition network (i.e., the encoder) regularized by the prior.

Traditionally, a simple prior, such as the standard normal distribution (Kingma & Welling, 2013), is used for tractability, simplicity, or not knowing what prior to use. It is hoped that this simple prior will be transformed somewhere in the deep generative network into a form suitable for characterizing the data distribution. While this might hold true when the generative network has enough capacity, applying the standard normal prior often results in over-regularized models with only few active latent dimensions (Burda et al., 2015).

Some recent works (Hoffman & Johnson, 2016; Goyal et al., 2017; Tomczak & Welling, 2017) suggest that the choice of the prior may have a profound impact on the expressiveness of the model. As an example, in learning the VAE with a simple encoder and decoder, Hoffman & Johnson (2016) conjecture that multimodal priors can achieve a higher variational lower bound on the data log-likelihood than is possible with the standard normal prior. Tomczak & Welling (2017) confirm the truth of this conjecture by showing that their multimodal prior, a mixture of the variational posteriors, consistently outperforms simple priors on a number of datasets in terms of maximizing the data log-likelihood. Taking one step further, Goyal et al. (2017) learn a tree-structured nonparametric Bayesian prior for capturing the hierarchy of semantics presented in the data. All these priors are learned under the VAE framework following the principle of maximum likelihood.

Along a similar line of thinking, we propose in this paper the notion of code generators for learning a prior from data for AAE. The objective is to learn a code generator network to transform a simple prior into one that, together with the generative network, can better characterize the data distribution. To this end, we generalize the framework of AAE in several significant ways:

- We replace the simple prior with a learned prior by training the code generator to output latent variables that will minimize an adversarial loss in data space.

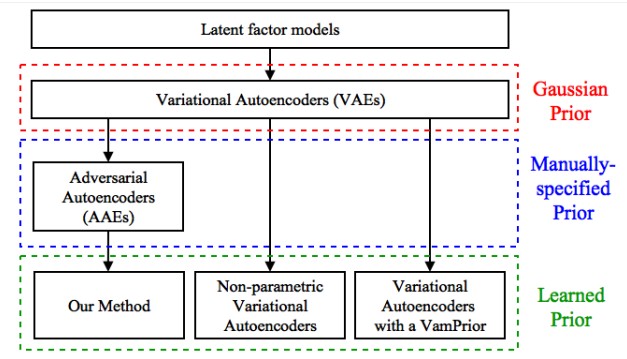

Figure 1: The relations of our work with prior arts.

- We employ a learned similarity metric (Larsen et al., 2015) in place of the default squared error in data space for training the autoencoder.
- We maximize the mutual information between part of the code generator input and the decoder output for supervised and unsupervised training using a variational technique introduced in InfoGAN (Chen et al., 2016).

Extensive experiments confirm its effectiveness of generating better quality images and learning better disentangled representations than AAE in both supervised and unsupervised settings, particularly on complicated datasets. In addition, to the best of our knowledge, this is one of the first few works that attempt to introduce a learned prior for AAE.

The remainder of this paper is organized as follows: Section 2 reviews the background and related works. Section 3 presents the implementation details and the training process of the proposed code generator. Section 4 compares its performance with AAE in image generation and disentanglement tasks. Lastly, we conclude this paper with remarks on future work.

## 2    BACKGROUND AND RELATED WORK

A latent factor model is a probabilistic model for describing the relationship between a set of latent and visible variables. The model is usually specified by a prior distribution $p(z)$ over the latent variables $z$ and a conditional distribution $p(x|z; \theta)$ of the visible variables $x$ given the latent variables $z$. The model parameters $\theta$ are often learned by maximizing the marginal log-likelihood of the data $\log p(x; \theta)$.

**Variational Autoencoders (VAEs).** To improve the model's expressiveness, it is common to make deep the conventional latent factor models by introducing a neural network to $p(x|z; \theta)$. One celebrated example is VAE (Kingma & Welling, 2013), which assumes the following prior $p(z)$ and $p(x|z; \theta)$:

$$p(z) \sim \mathcal{N}(z; 0, I)$$
$$p(x|z; \theta) \sim \mathcal{N}(x; o(z; \theta), \sigma^2 I) \tag{1}$$

where the mean $o(z; \theta)$ is modeled by the output of a neural network with parameters $\theta$. In this case, the marginal $p(x; \theta)$ becomes intractable; the model is thus trained by maximizing the log evidence lower-bound (ELBO):

$$\mathcal{L}(\phi, \theta) = E_{q(z|x; \phi)} \log p(x|z; \theta) - KL(q(z|x; \phi) \parallel p(z)) \tag{2}$$

where $q(z|x; \phi)$ is the variational density, implemented by another neural network with parameter $\phi$, to approximate the posterior $p(z|x; \theta)$. When regarding $q(z|x; \phi)$ as an (stochastic) encoder and $p(z|x; \theta)$ as a (stochastic) decoder, Equation (2) bears an interpretation of training an autoencoder with the latent code $z$ regularized by the prior $p(z)$ through the KL-divergence.

**Adversarial Autoencoders (AAEs).** Motivated by the observation that VAE is largely limited by the Gaussian prior assumption, i.e., $p(z) \sim \mathcal{N}(z; 0, I)$, Makhzani et al. (2015) relax this constraint

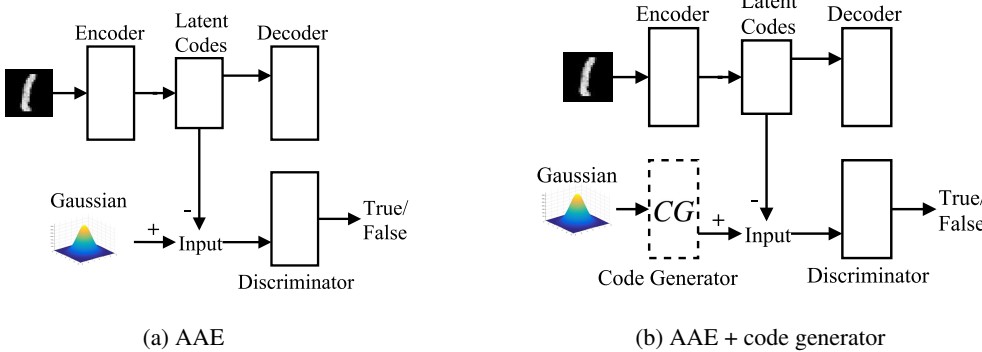

(a) AAE

(b) AAE + code generator

Figure 2: The architecture of AAE without (left) and with (right) the code generator.

by allowing $p(z)$ to be any distribution. Apparently, the KL-divergence becomes intractable when $p(z)$ is arbitrary. They thus replace the the KL-divergence with an adversarial loss imposed on the encoder output, requiring that the latent code $z$ produced by the encoder should have an aggregated posterior distribution[1] the same as the prior $p(z)$.

**Non-parametric Variational Autoencoders (Non-parametric VAEs).** While AAE allows the prior to be arbitrary, how to select a prior that can best characterize the data distribution remains an open issue. Goyal et al. (2017) make an attempt to learn a non-parametric prior based on the nested Chinese restaurant process for VAEs. Learning is achieved by fitting it to the aggregated posterior distribution, which amounts to maximization of ELBO. The result induces a hierarchical structure of semantic concepts in latent space.

**Variational Mixture of Posteriors (VampPrior).** The VampPrior is a new type of prior for the VAE. It consists of a mixture of the variational posteriors conditioned on a set of learned pseudo-inputs $\{x_k\}$. In symbol, this prior is given by

$$p(z) = \frac{1}{K} \sum_{k=1}^{K} q(z|x_k; \phi) \tag{3}$$

Its multimodal nature and coupling with the posterior achieve superiority over many other simple priors in terms of training complexity and expressiveness.

Inspired by these learned priors (Goyal et al., 2017; Tomczak & Welling, 2017) for VAE, we propose in this paper the notion of code generator to learn a proper prior from data for AAE. The relations of our work with these prior arts are illustrated in Fig. 1.

## 3 LEARNING THE PRIOR

In this paper, we propose to learn the prior from data instead of specifying it arbitrarily. Built on the foundation of AAE, we introduce a neural network (which we call the *code generator*) to transform the manually-specified prior into a better form. Fig. 2 presents its role in the overall architecture, and contrasts the architectural difference relative to AAE.

Because this code generator itself has to be learned, we need an objective function to shape the distribution at its output. Normally, we wish to find a prior that, together with the decoder in Fig. 3, would lead to a distribution that maximizes the data likelihood. We are however faced with two challenges. First, the output of the code generator could be any distribution, which makes the likelihood function and its variational lower bound intractable. Second, the decoder has to be learned simultaneously, which creates a moving target for the code generator.

To address the first challenge, we propose to impose an adversarial loss on the output of the decoder when training the code generator. That is, we want the code generator to produce a prior that minimizes the adversarial loss at the decoder output. Using the example in Fig. 4, the decoder

---

[1]The aggregated posterior distribution is defined as $q(z) = \int q(z|x; \phi) p_d(x) dx$, where $p_d(x)$ denotes the empirical distribution of the training data

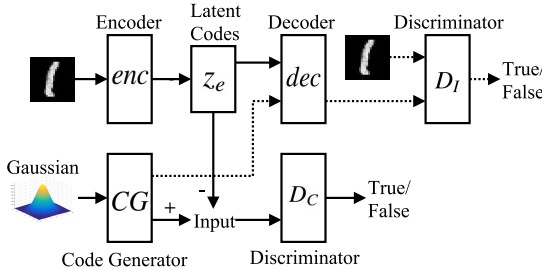

Figure 3: The overall training architecture.

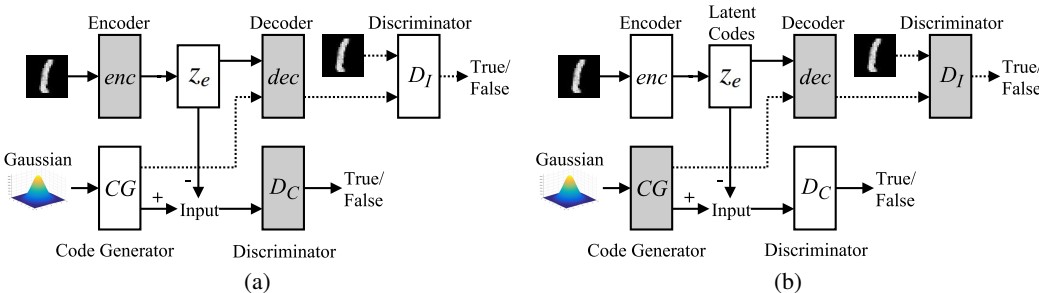

Figure 4: Alternation of training phases: (a) the AAE phase and (b) the prior improvement phase.

should generate images with a distribution that in principle matches the empirical distribution of real images in the training data, when driven by the output samples from the code generator. In symbols, this is to minimize

$$\mathcal{L}^I_{GAN} = \log(D_I(x)) + \log(1 - D_I(dec(z_c))), \tag{4}$$

where $z_c = CG(z)$ is the output of the code generator $CG$ driven by a noise sample $z \sim p(z)$, $D_I$ is the discriminator in image space, and $dec(z_c)$ is the output of the decoder driven by $z_c$.

To address the second challenge, we propose to alternate training of the code generator and the decoder/encoder until convergence. In one phase, termed the *prior improvement phase*. we update the code generator with the loss function in Eq. (4), by fixing the encoder[2]. In the other phase, termed the *AAE phase*, we fix the code generator and update the autoencoder following the training procedure of AAE. Specifically, the encoder output has to be regularized by the following adversarial loss:

$$\mathcal{L}^C_{GAN} = \log(D_C(z_c)) + \log(1 - D_C(enc(x))), \tag{5}$$

where $z_c = CG(z)$ is the output of the code generator, $enc(x)$ is the encoder output given the input $x$, and $D_C$ is the discriminator in latent code space.

Because the decoder will be updated in both phases, the convergence of the decoder relies on consistent training objectives during the alternation of training phases. It is however noticed that the widely used pixel-wise squared error criterion in the AAE phase tends to produce blurry decoded images. This obviously conflicts with the adversarial objective in the prior improvement phase, which wants the decoder to produce sharp images. Inspired by the notion of learning similarity metrics (Larsen et al., 2015), we change the criterion of minimizing squared error in pixel domain to be in feature domain. Specifically, in the AAE phase, we require that a decoded image $dec(enc(x))$ should minimize the squared error $\|\mathcal{F}(dec(enc(x))) - \mathcal{F}(x)\|^2$ with the input image $x$ in feature domain, where $\mathcal{F}(\cdot)$ denotes the feature representation of an image (usually the output of the last convolutional layer) in the image discriminator $D_I$. With this, the decoder would be driven consistently in both phases towards producing decoded images that resemble closely real images.

---

[2]Supposedly, the decoder needs to be fixed in this phase. It is however found beneficial in terms of convergence to update also the decoder.

---

**Algorithm 1** Training algorithm for our method.

---

$\theta_{enc}, \theta_{dec}, \theta_{CG}, \theta_{D_I}, \theta_{D_C}, \theta_Q \longleftarrow$ Initialize network parameters
Repeat (for each epochs $E_i$)
   Repeat (for each mini-batch $x_j$)
     // AAE phase
     $z \sim p(z)$
     If conditional variables $s$ exist then
         $z_c \leftarrow CG(z, s)$
     Else
         $z_c \leftarrow CG(z)$
     End If

     $\mathcal{L}_{GAN}^C \leftarrow \log(D_C(z_c)) + \log(1 - D_C(enc(x)))$
     $x_{rec} \leftarrow dec(enc(x))$
     $\mathcal{L}_{rec} \leftarrow \frac{1}{N}\|\mathcal{F}(x) - \mathcal{F}(x_{rec})\|_2$

     // Update network parameters for AAE phase
     $\theta_{D_C} \leftarrow \theta_{D_C} - \nabla_{\theta_{D_C}}(\mathcal{L}_{GAN}^C)$
     $\theta_{enc} \leftarrow \theta_{enc} - \nabla_{\theta_{enc}}(-\mathcal{L}_{GAN}^C + \mathcal{L}_{rec})$
     $\theta_{dec} \leftarrow \theta_{dec} - \nabla_{\theta_{dec}}(\lambda * \mathcal{L}_{rec})$

     // Prior improvement phase
     $z \sim p(z)$
     If conditional variables $s$ exist then
         $z_c \leftarrow CG(z, s)$
     Else
         $z_c \leftarrow CG(z)$
     End If

     $x_{noise} \leftarrow dec(z_c)$
     $x_{rec} \leftarrow dec(enc(x_j))$
     $\mathcal{L}_{GAN}^I \leftarrow \log(D_I(x_j)) + \log(1 - D_I(x_{noise})) + \log(1 - D_I(x_{rec}))$

     // Update network parameters for prior improvement phase
     $\theta_{D_I} \leftarrow \theta_{D_I} - \nabla_{\theta_{D_I}}(\mathcal{L}_{GAN}^I)$
     If conditional variables $s$ exist then
         $\theta_{dec} \leftarrow \theta_{dec} - \nabla_{\theta_{dec}}(-\mathcal{L}_{GAN}^I + I(s; dec(z_c)))$
         $\theta_Q \leftarrow \theta_Q - \nabla_{\theta_Q}(I(s; dec(z_c)))$
     Else
         $\theta_{dec} \leftarrow \theta_{dec} - \nabla_{\theta_{dec}}(-\mathcal{L}_{GAN}^I)$
     End If
   Until all mini-batches are seen
Until terminate

---

Moreover, when it is necessary to generate images conditionally on an input variable $s$ to the code generator, as will be seen in our supervised and unsupervised learning tasks, we introduce the variational learning technique in InfoGAN (Chen et al., 2016) to maximize the mutual information $I(s; dec(z_c))$ between the variable $s$ and the generated image. This way we explicitly force the code generate to pick up the information carried by the variable $s$ when generating the latent code.

## 4 EXPERIMENTS

We compare the performance of our model with AAE, which adopts manually-specified priors, in image generation and disentanglement tasks. In Section 4.1, we show that using the same encoder and decoder architecture, our model with code generator and similarity metric learning can generate higher quality images. In Section 4.2, we demonstrate that our model can better learn disentan-

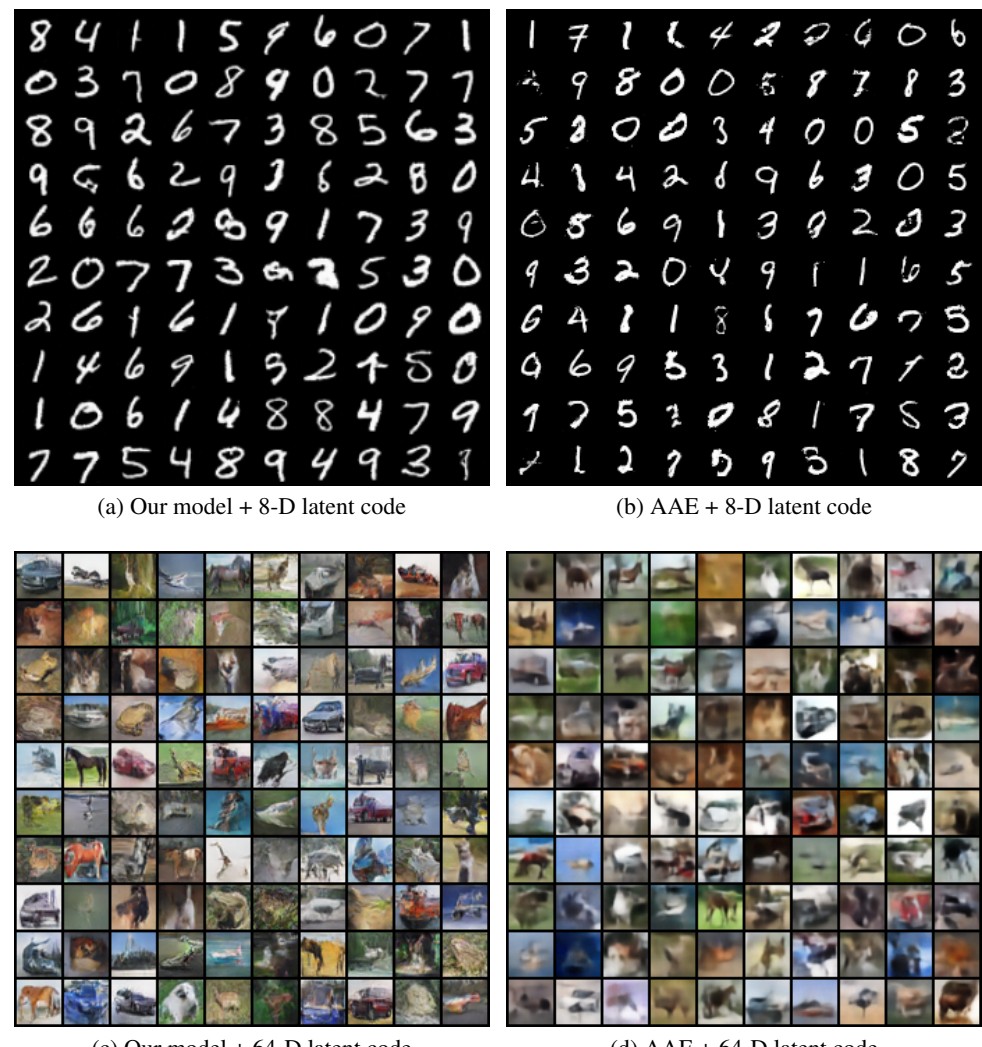

(a) Our model + 8-D latent code           (b) AAE + 8-D latent code

(c) Our model + 64-D latent code         (d) AAE + 64-D latent code

Figure 5: Images generated by our model and AAE trained on MNIST (upper) and CIFAR-10 (lower).

gled representations in both supervised and unsupervised settings. In Section 4.3, we present an application of our model to text-to-image synthesis.

## 4.1 IMAGE GENERATION

Latent factor models with the priors learned from data rather than specified arbitrarily should ideally better characterize the data distribution. To verify this, we compare the performance of our model with AAE (Makhzani et al., 2015), in terms of image generation. In this experiment, the autoencoder in our model is trained based on minimizing the squared reconstruction error in feature domain (i.e., the learned similarity metric), whereas by convention, AAE is trained by minimizing the squared error in data domain. For a fair comparison, we require that both models have access to the same encoder and decoder networks, with the network parameters trained to optimize their respective priors.

Fig. 5 displays side-by-side images generated from these models when trained on MNIST and CIFAR-10 datasets. They are produced by drawing samples from the priors and passing them through their respective decoders. In this experiment, two observations are immediate. First, our

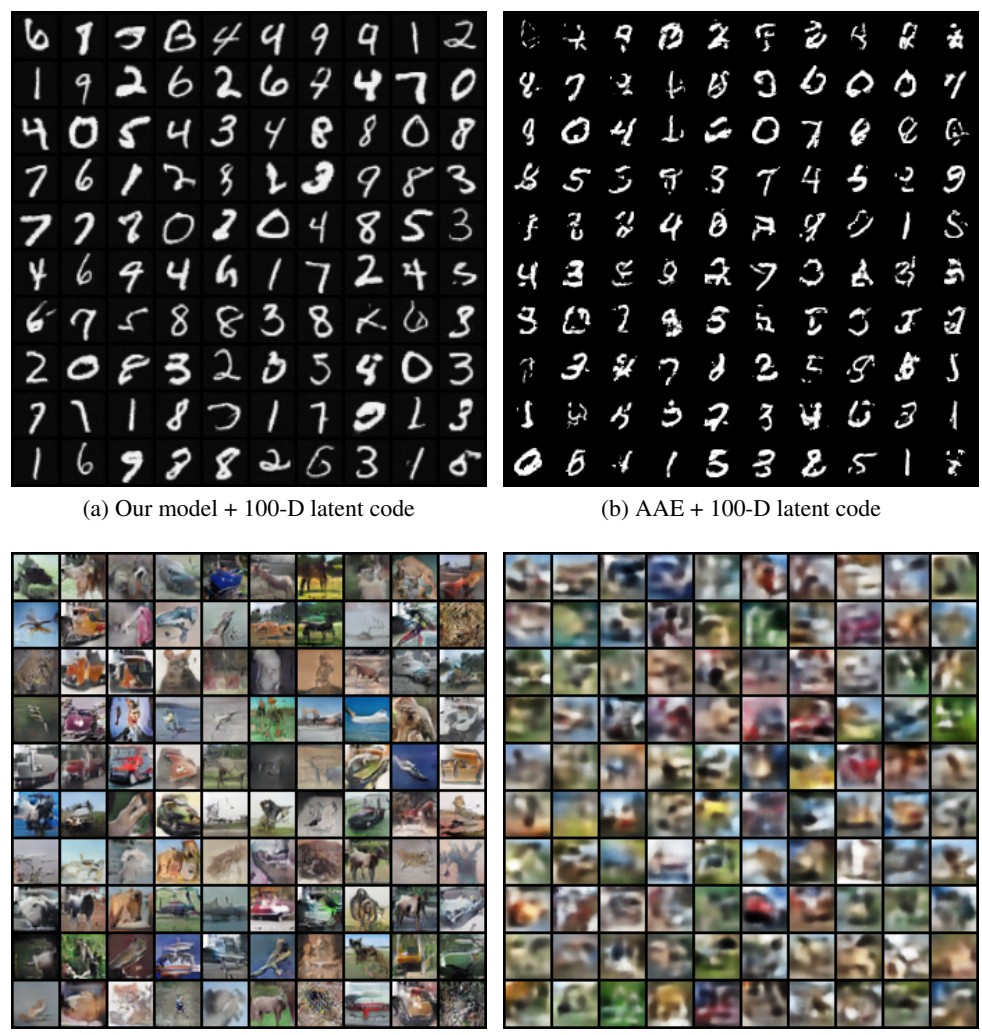

(a) Our model + 100-D latent code

(b) AAE + 100-D latent code

(c) Our model + 2000-D latent code

(d) AAE + 2000-D latent code

Figure 6: Images generated by our model and AAE trained on MNIST (upper) and CIFAR-10 (lower). In this experiment, the latent code dimension is increased significantly to 64-D and 2000-D for MNIST and CIFAR-10, respectively. For AAE, the re-parameterization trick is applied to the output of the encoder as suggested in (Makhzani et al., 2015).

Table 1: Inception score of different generative models on CIFAR-10

| Method | Inception Score |
|---|---|
| DCGAN | 6.16 |
| WGAN-GP | 7.86 |
| BEGAN | 5.62 |
| DFM | 7.72 |
| Our method w/ a learned prior | 6.52 |
| Our method w/ a Gaussian prior | 6.02 |

model can generate sharper images than AAE on both datasets. Second, AAE experiences problems in reconstructing visually-plausible images on the more complicated CIFAR-10. These highlight the advantage of optimizing with respect to a learned similarity metric and learning the code generator through an adversarial loss, which in general produces subjectively sharper images. Table 1 compares the inception score of our model with some other generative models on CIFAR-10. Caution must be exercised in interpreting these numbers as they implement different generative networks. With the current implementation, our model achieves a comparable score to other generative models. Moreover, the use of a learned prior does not improve further on generation quality.

Another advantage of our model is its ability to have better adaptability in high-dimensional latent code space. Fig. 6 presents images generated by the two models when the dimension of the latent code is increased significantly from 8 to 100 on MNIST, and from 64 to 2000 on CIFAR-10. As compared to Fig. 5, it is seen that the increase in code dimension has little impact on our model, but exerts a strong influence on AAE. In the present case, AAE can hardly produce recognizable images, particularly on CIFAR-10, even after the re-parameterization trick has been applied to the output of the encoder as suggested in (Makhzani et al., 2015). This emphasizes the importance of having a prior that can adapt automatically to changes in code space and data.

## 4.2 DISENTANGLED REPRESENTATION

Learning disentangled representation is desirable in many applications. It refers generally to learning a representation whose individual dimensions can capture independent factors of variation in the data. To demonstrate the ability of our model to learn disentangled representations and the merits of GAN-driven priors, we repeat the disentanglement tasks in (Makhzani et al., 2015), and compare its performance with AAE.

### 4.2.1 SUPERVISED LEARNING

This session presents experimental results of a network architecture that incorporates the GAN-driven prior in learning supervisedly to disentangle the label information of images from the remaining information. Its block diagram is depicted in Fig. 7, where the code generator takes as input the label information of an image and an independent Gaussian noise to impose a conditional latent code distribution on the image representation. This has an interpretation of associating each class of images with a code space governed by some distribution conditioned on the label. In particular, this conditional distribution itself needs to be learned from data using the GAN-based training procedure presented in Figure 7. To enforce the use of the label information for image generation, we additionally apply the variational technique proposed in (Chen et al., 2016) to maximize the mutual information between the label and the generated image. At test time, image generation for a particular class is achieved by inputting the class label and a Gaussian noise to the code generator and then passing the resulting code through the decoder. Particularly, to see the sole contribution from the learned prior, the AAE baseline also adopts a learned similarity metric and this same mutual information maximization; that is, the only difference relative to our model is the use of a manually-specified prior governed by a one-hot vector and a standard normal noise.

Fig. 8 displays images generated by our model and AAE. Both models adopt a 10-D one-hot vector to specify the label and a 54-D Gaussian to generate the noise. To be fair, the output of our code generator has an identical dimension (i.e., 64) to the latent prior of AAE. Each row of Fig. 8 corresponds to images generated by varying the label while fixing the noise. Likewise, each column shows images that share the same label yet with varied noise.

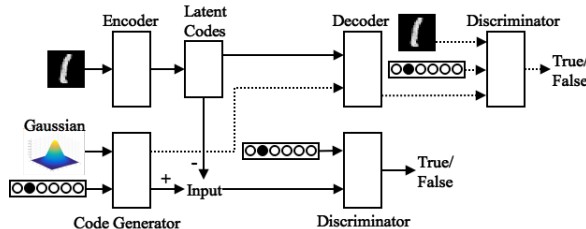

Figure 7: Supervised learning architecture with the code generator.

On MNIST and SVHN, both models work well in separating the label information from the remaining (style) information. This is evidenced from the observation that along each row, the main digit changes with the label input regardless of the noise variable, and that along each column, the style varies without changing the main digit. On CIFAR-10, the two models behave differently. While both produce visually plausible images, ours generate more semantically discernible images that match the labels.

Fig. 9 visualizes the output of the code generator with the t-distributed stochastic neighbor embedding (t-SNE). It is seen that the code generator learns a distinct conditional distribution for each class of images. It is believed that the more apparent inter-class distinction reflects the more difficult it is for the decoder to generate images of different classes. Moreover, the elliptic shape of the intra-class distributions in CIFAR-10 may be ascribed to the higher intra-class variability.

### 4.2.2 UNSUPERVISED LEARNING

This session presents experimental results of our model in learning unsupervisedly to disentangle the label information of images from the remaining information. As illustrated in Fig. 10, this is achieved by dividing the input to the code generator into two parts, one driven by an uniform categorial distribution and the other by a Gaussian. The categorical distribution encodes our prior belief about data clusters. The number of distinct values over which it is defined specifies the presumed number of clusters in the data. The Gaussian serves to explain the data variability within each cluster. These two distributions are further mingled together by the fully connected layers in the code generator, to form a prior that is best suited for explaining the data. Again, the AAE baseline differs by the use of a manually-specified prior.

At test time, image generation is done similarly to the supervised case. We start by sampling the categorical and Gaussian distributions, followed by feeding the samples into the code generator and then onwards to the decoder. In this experiment, the categorical distribution is defined over 10-D one-hot vectors, which denote the label variable, and the Gaussian is 90-D. As in the supervised setting, after the model is trained, we alter the label variable or the Gaussian noise one at a time to verify whether the model has learned to cluster images. We expect that a good model should generate images of the same digit or of the same class when the Gaussian part is altered while the label part remains fixed.

The results in Fig. 11 show that on MNIST, both our model and AAE successfully learn to disentangle the label from the remaining information. Based on the same presentation order as in the supervised setting, we see that each column of images (which correspond to the same label variable) do show images of the same digit. This is however not the case on the more complicated SVHN and CIFAR-10 datasets: each column could mix images from different digits/classes. Nevertheless, both models have a tendency to cluster images with similar background colors.

Fig. 12 further visualizes the latent code distributions at the output of the code generator and the encoder. Several observations can be made. First, the encoder is regularized well to produce an aggregated posterior distribution similar to that at the code generator output. Second, the code generator learns distinct conditional distributions according to the categorical label input. Third, the encoder successfully learns to cluster images of the same digit on MNIST, as has been confirmed in Fig. 11. As expected, such clustering phenomenon in code space is not obvious on SVHN and

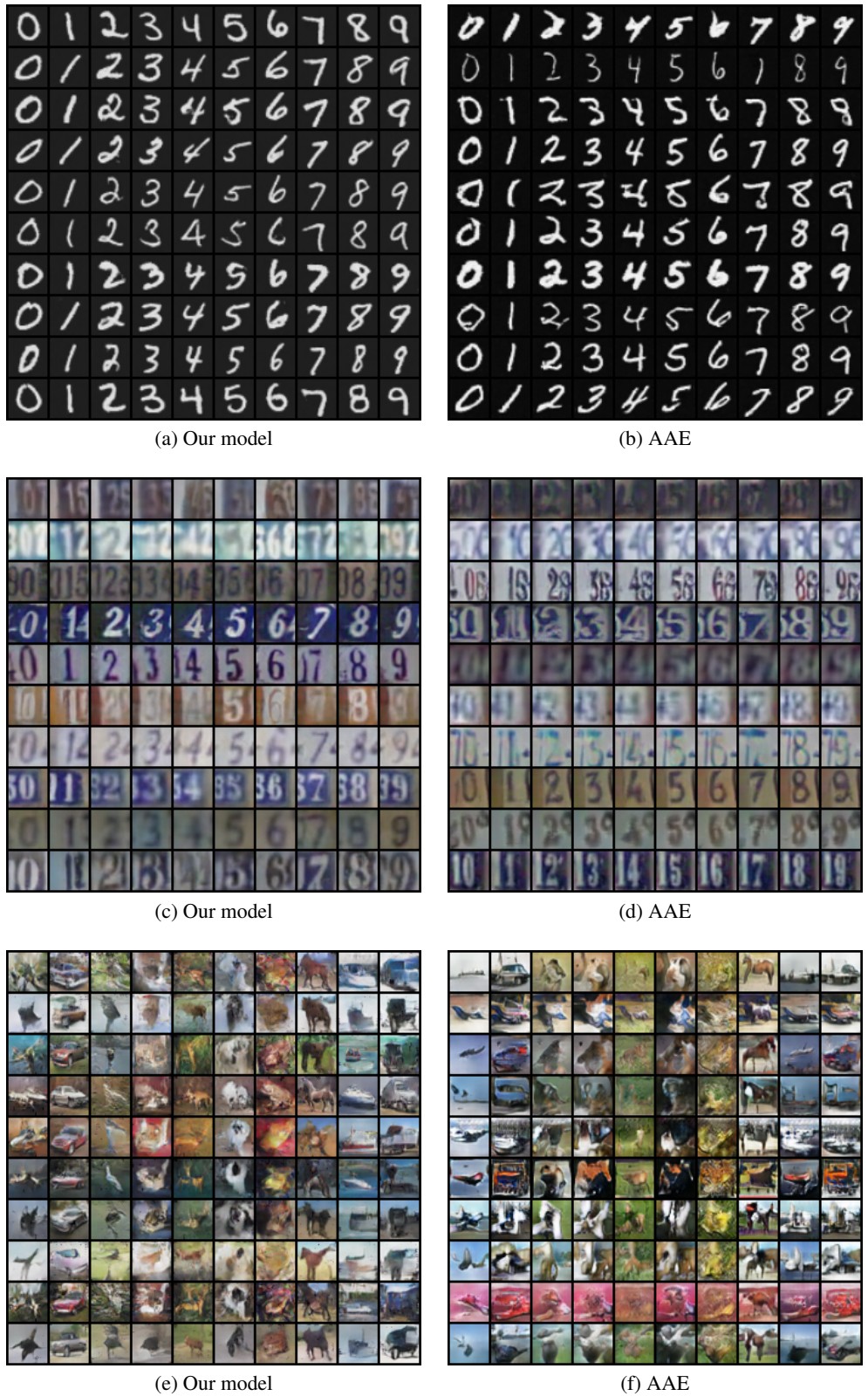

Figure 8: Images generated by the proposed model (a)(c)(e) and AAE (b)(d)(f) trained on MNIST, SVHN and CIFAR-10 datasets in the supervised setting. Each column of images have the same label/class information but varied Gaussian noise. On the other hand, each row of images have the same Gaussian noise but varied label/class variables.

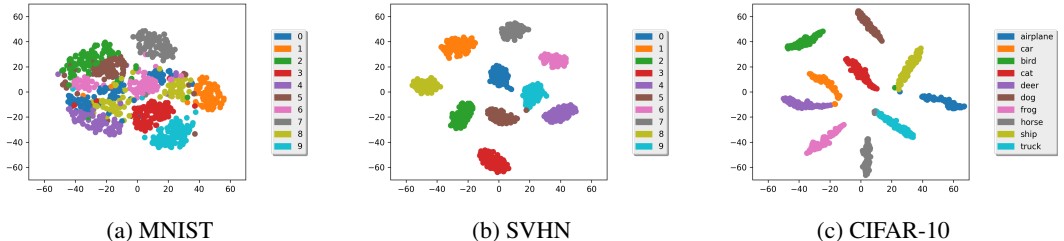

(a) MNIST          (b) SVHN          (c) CIFAR-10

Figure 9: Visualization of the code generator output in the supervised setting.

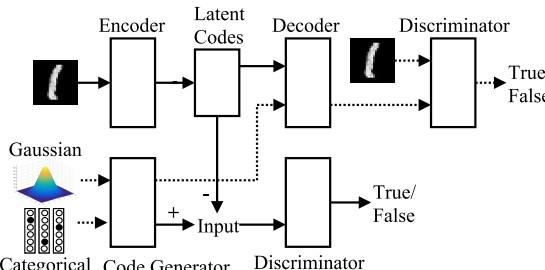

Figure 10: Unsupervised learning architecture with the code generator.

CIFAR-10, as is evident from the somewhat random assignment of latent codes to images of the same class.

### 4.3 TEXT-TO-IMAGE SYNTHESIS

This session presents an application of our model to text-to-image synthesis. We show that the code generator can transform the embedding of a sentence into a prior suitable for synthesizing images that match closely the sentence's semantics. To this end, we learn supervisedly the correspondence between images and their descriptive sentences using the architecture in Fig. 7, where given an image-sentence pair, the sentence's embedding (which is a 200-D vector) generated by a pre-trained recurrent neural network is input to the code generator and the discriminator in image space as if it were the label information, while the image representation is learned through the autoencoder and regularized by the output of the code generator. As before, a 100-D Gaussian is placed at the input of the code generator to explain the variability of images given the sentence.

The results in Fig. 13 present images generated by our model when trained on 102 Category Flower dataset (Nilsback & Zisserman, 2008). The generation process is much the same as that described in Section 4.2.1. It is seen that most images match reasonably the text descriptions. In Fig. 14, we further explore how the generated images change with the variation of the color attribute in the text description. We see that most images agree with the text descriptions to a large degree.

## 5 CONCLUSION

In this paper, we propose to learn a proper prior from data for AAE. Built on the foundation of AAE, we introduce a code generator to transform the manually selected simple prior into one that can better fit the data distribution. We develop a training process that allows to learn both the autoencoder and the code generator simultaneously. We demonstrate its superior performance over AAE in image generation and learning disentangled representations in supervised and unsupervised settings. We also show its ability to do cross-domain translation. Mode collapse and training instability are two major issues to be further investigated in future work.

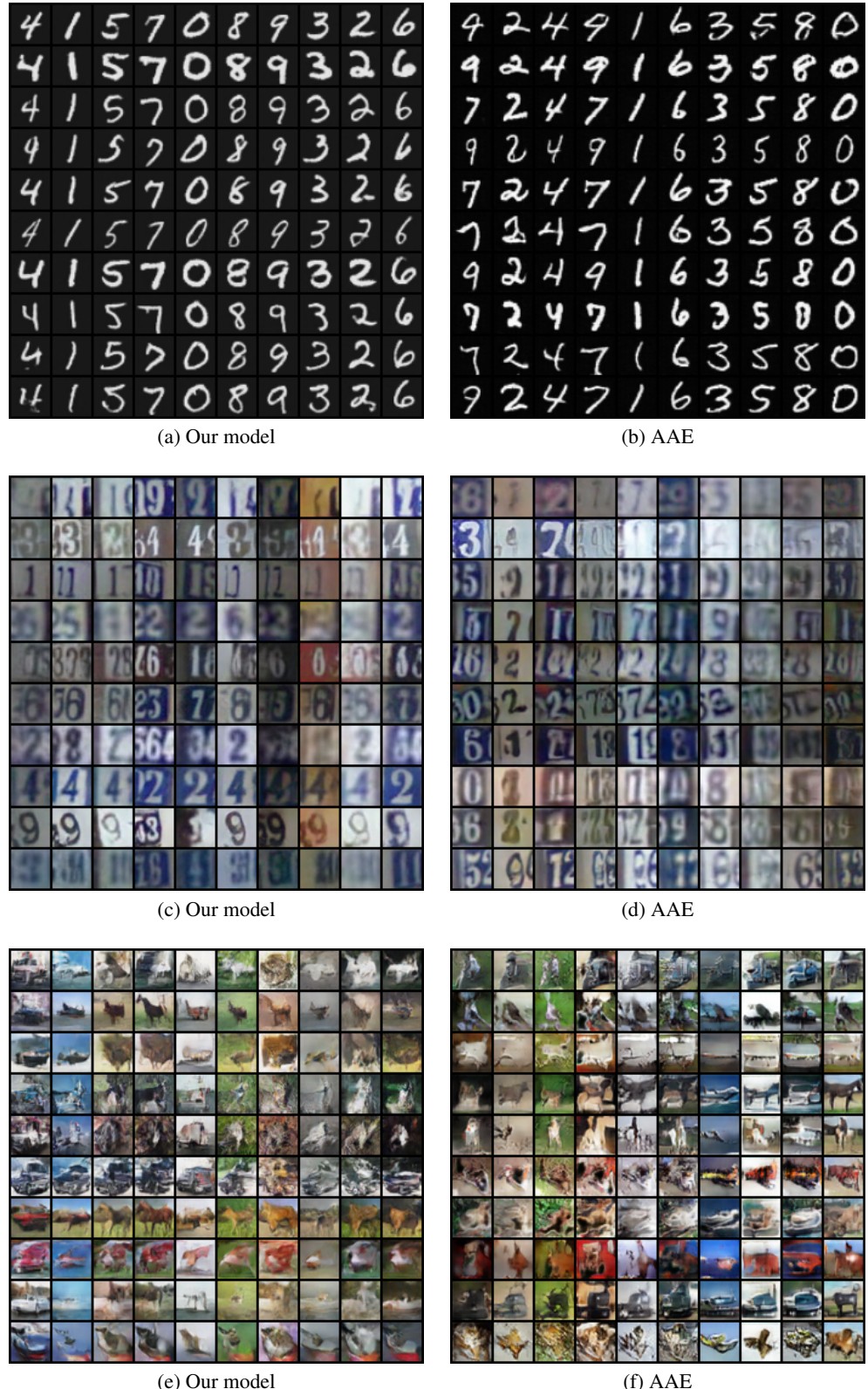

Figure 11: Images generated by the proposed model (a)(c)(e) and AAE (b)(d)(f) trained on MNIST, SVHN and CIFAR-10 datasets in the unsupervised setting. Each column of images have the same label/class information but varied Gaussian noise. On the other hand, each row of images have the same Gaussian noise but varied label/class variables.

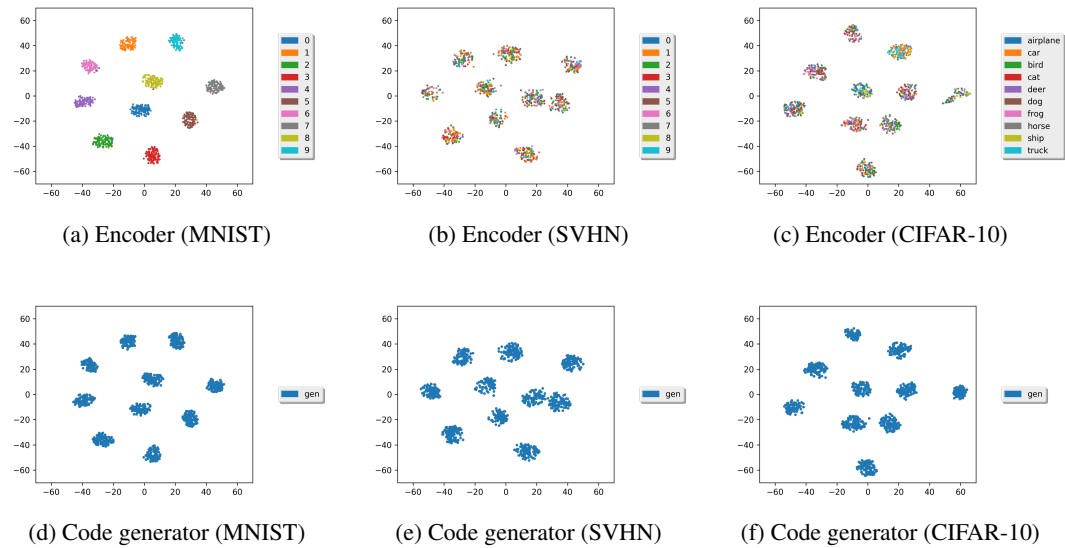

(a) Encoder (MNIST)      (b) Encoder (SVHN)      (c) Encoder (CIFAR-10)

(d) Code generator (MNIST)      (e) Code generator (SVHN)      (f) Code generator (CIFAR-10)

Figure 12: Visualization of the encoder output versus the code generator output in the unsupervised setting.

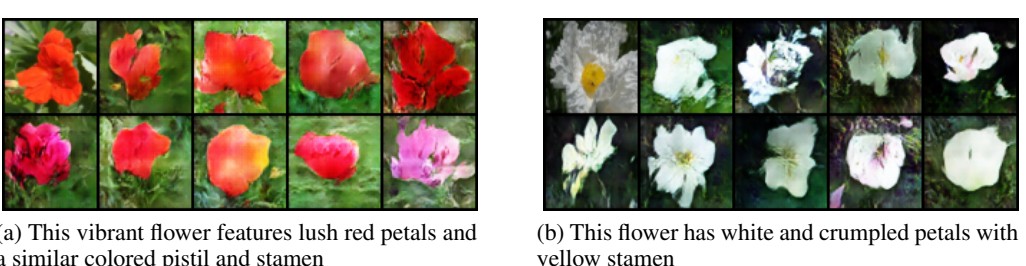

(a) This vibrant flower features lush red petals and a similar colored pistil and stamen

(b) This flower has white and crumpled petals with yellow stamen

Figure 13: Generated images from text descriptions.

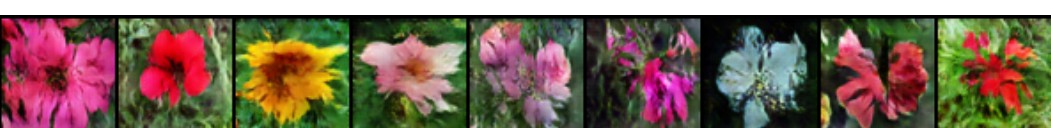

Figure 14: Generated images in accordance with the varying color attribute in the text description "The flower is pink in color and has petals that are rounded in shape and ruffled." From left to right, the color attribute is set to pink, red, yellow, orange, purple, blue, white, green, and black, respectively. Note that there is no green or black flower in the dataset.

Table 2: Implementation details of the encoder and decoder networks

| Encoder | Decoder |
|---|---|
| Input 32 x 32 images | Input latent code $\in R^{code\ size}$ |
| 3 x 3 conv. 64 RELU stride 2 pad 1 | 4 x 4 upconv. 512 BN. RELU stride 1 |
| 3 x 3 residual blcok 64 | 4 x 4 up sampling residual block 256 stride 2 |
| 3 x 3 down sampling residual blcok 128 stride 2 | 4 x 4 up sampling residual block 128 stride 2 |
| 3 x 3 down sampling residual blcok 256 stride 2 | 4 x 4 up sampling residual block 64 stride 2 |
| 3 x 3 down sampling residual block 512 stride 2 | 3 x 3 conv. image channels Tanh |
| 4 x 4 avg. pooling stride 1 | |
| FC. 2 x code size BN. RELU | |
| FC. code size Linear | |

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

## APPENDIX

Table 2, Table 3, and Table 4 presents the implementation details of each components in our model. Each cell in the tables presents the type of neural networks, the output size, w/o batch normalization, the type of activation function, the size for strides, and the size of padding. Lastly, Fig. 15 presents the detailed architecture of the proposed model.

Table 3: Implementation details of the code generator networks

| Code Generator | Residual block |
|---|---|
| Input noise $\in R^{noise\ size}$ | Input feature map |
| FC. 2 x noise size BN. RELU | 3 x 3 conv. out_channels RELU stride 2 pad 1 |
| FC. latent code size BN. Linear | 3 x 3 conv. out_channels RELU stride 1 pad 1 |
| | skip connection output = input + residual |
| | RELU |

Table 4: Implementation details of the image and code discriminator

| Image Discriminator $D$/$Q$ | Code Discriminator |
|---|---|
| Input 32 x 32 images | Input latent code |
| 4 x 4 conv. 64 LRELU stride 2 pad 1 | FC 1000 LRELU |
| 4 x 4 conv. 128 BN LRELU stride 2 pad 1 | FC 500 LRELU |
| 4 x 4 conv. 256 BN LRELU stride 2 pad 1 | FC 200 LRELU |
| FC. 1000 LRELU | FC 1 Sigmoid |
| FC 1 Sigmoid for $D$ | |
| FC 10 Softmax for $Q$ | |

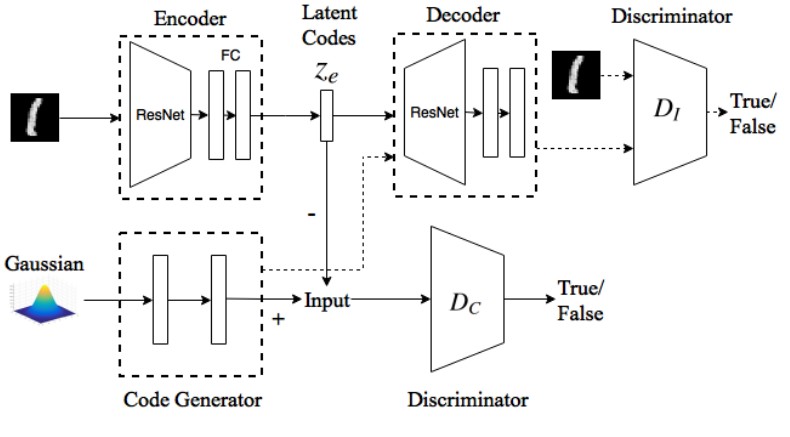

Figure 15: The detailed model architecture.

