# OpenReview forum: "Learning Priors for Adversarial Autoencoders"
_ICLR.cc/2018/Conference — Reject_

### Official Review · AnonReviewer1 · 2017-11-27
**Interesting idea, but more thorough analysis is needed.**

**Rating:** 6
**Confidence:** 3

**Review:**

This paper proposes an interesting idea--to learn a flexible prior from data by maximizing data likelihood.

It seems that in the prior improvement stage, what you do is training a GAN with CG+dec as the generator while D_I as the discriminator (since you also update dec at the prior improvement stage). So it can also be regarded as GAN trained with an additional enc and D_c, and additional objective. In my opinion, this may explain why your model can generate sharper images.

The experiments do demonstrate the power of their model compared to AAE. However, only the qualitative analysis may not persuade me and more thorough analysis is needed.

1. About the latent space for z. The motivation in AAE is to impose aggregated posterior regularization $D(q(z),p(z))$ where $p(z)$ is chosen as a simple one, e.g., Gaussian. I'm curious how the geometry of the latent space will be, when the code generator is introduced. Maybe some visualization like t-sne will be helpful.
2. Any quantitative analysis? Doing a likelihood analysis like that in the AAE paper will be very informative.

---

> ### Author Response · Authors · 2018-01-06
> **Response to Reviewer 1**
>
> Dear Reviewer 1,
>
> "This paper proposes an interesting idea--to learn a flexible prior from data by maximizing data likelihood. It seems that in the prior improvement stage, what you do is training a GAN with CG+dec as the generator while D_I as the discriminator (since you also update dec at the prior improvement stage). So it can also be regarded as GAN trained with an additional enc and D_c, and additional objective. In my opinion, this may explain why your model can generate sharper images.
> The experiments do demonstrate the power of their model compared to AAE. However, only the qualitative analysis may not persuade me and more thorough analysis is needed. "
>
> Thanks for your suggestions. We have provided more analysis results including comparison of inception scores and visualization of learned code space in the revised manuscript.
>
> "1. About the latent space for z. The motivation in AAE is to impose aggregated posterior regularization $D(q(z),p(z))$ where $p(z)$ is chosen as a simple one, e.g., Gaussian. I'm curious how the geometry of the latent space will be, when the code generator is introduced. Maybe some visualization like t-sne will be helpful.
>
> 2. Any quantitative analysis? Doing a likelihood analysis like that in the AAE paper will be very informative. "
>
> Thanks for your suggestion. For quantitative evaluation, we have compared the inception score of the proposed method with other generative models in Table I. We also have visualized the learned priors with t-SNE in Figs. 9 and 12 for the supervised and unsupervised learning tasks. The text in Section 4.2.1 and Section 4.2.2 have been modified accordingly to include the discussions (see the last paragraphs in these sections).
>
> In addition, since receiving the review comments, we have improved our model in several significant ways, including
> 1) Introducing a pair of more capable encoder and decoder with ResNets. (See appendix for the implementation details)
> 2) Employing a learned similarity metric in place of the default squared error in data space to improve the convergence of the decoder. (See Section 3 Learning The Prior for the reasons)
> 3) Introducing the variational technique in InfoGAN for training the decoder and code generator when it is necessary to generate images conditionally on an input variable, as in our supervised and unsupervised learning tasks. (See Section 3 Learning The Prior for the reasons) With these changes, our model can now produce much better images without incurring obvious mode collapse.
>
> We have re-written extensively the entire manuscript, presenting more experimental results and analyses as requested.

---

### Official Review · AnonReviewer3 · 2017-11-28
**Improving AAE by warping the Gaussian prior using deep networks**

**Rating:** 5
**Confidence:** 3

**Review:**

This paper propose a simple extension of the adversarial auto-encoders for (conditional) image generation. The general idea is that instead of using Gaussian prior, the propose algorithm uses a "code generator" network  to warp the gaussian distribution, such that the internal prior of the latent encoding space is more expressive and complicated.

Pros:
- The proposed idea is simple and easy to implement
- The results show improvement in terms of visual quality

Cons:
- I agree that the proposed prior should better capture the data distribution. However, incorporating a generic prior over the latent space plays a vital role as regularisation, this helps avoid model collapse. Adding a complicated code generation network brings too much flexibility for the prior part. This makes the prior and posterior learnable, which makes it easier to fool the regularisation discriminator (think about the latent code and prior code collapsed to two different points). As a result, this weakens the regularisation over the latent encoder space.
- The above mentioned could be verified through qualitative results. As shown in Fig. 5. I believe this is a result due to the fact that the adversarial loss in the regularisation phase does not a significant influence there.
- I have some doubts over why AAE works so poorly when the latent dimension is 2000. How to make sure it's not a problem of implementation or the model wasn't trapped into a bad local optima / saddle points. Could you justify this?
- Contributions; this paper propose an improvement over a existing model. However, neither the idea/insights it brought can be applied onto other generative models, nor the improvement bring a significant improvement over the-state-of-the-arts. I am wondering what the community will learn from this paper, or what the author would like to claim as significant contributions.

---

> ### Author Response · Authors · 2018-01-06
> **Response to Reviewer 3**
>
> Dear Reviewer 3,
>
> "This paper propose a simple extension of the adversarial auto-encoders for (conditional) image generation. The general idea is that instead of using Gaussian prior, the propose algorithm uses a "code generator" network to warp the gaussian distribution, such that the internal prior of the latent encoding space is more expressive and complicated.
>
> Pros:
> - The proposed idea is simple and easy to implement
> - The results show improvement in terms of visual quality
>
> Cons:
> - I agree that the proposed prior should better capture the data distribution. However, incorporating a generic prior over the latent space plays a vital role as regularisation, this helps avoid model collapse.
> Adding a complicated code generation network brings too much flexibility for the prior part. This makes the prior and posterior learnable, which makes it easier to fool the regularisation discriminator (think about the latent code and prior code collapsed to two different points). As a result, this weakens the regularisation over the latent encoder space.
> - The above mentioned could be verified through qualitative results. As shown in Fig. 5. I believe this is a result due to the fact that the adversarial loss in the regularisation phase does not a significant influence there. "
>
> Thanks for your comments. I agree that generic priors may help avoid mode collapse. However, it also risks overly regularizing the model, consequently decreasing its expressiveness.
>
> This work, like few other similar attempts for VAE, aims to learn a prior through a code generation network so that the resulting model can better explain the data distribution. Unlike the prior works, which are mostly based on maximizing the data log-likelihood, ours tries to learn the prior by minimizing an adversarial loss in data space.
>
> Since receiving the review comments, we have improved our model in several significant ways, including
> 1)	Introducing a pair of more capable encoder and decoder with ResNets. (See appendix for the implementation details)
> 2)	Employing a learned similarity metric in place of the default squared error in data space to improve the convergence of the decoder. (See Section 3 Learning The prior for the reasons)
> 3)	Introducing the variational technique in InfoGAN for training the decoder and code generator when it is necessary to generate images conditionally on an input variable, as in our supervised and unsupervised learning tasks. (See Section 3 Learning The Prior for the reasons)
>
> With these changes, our model can now produce much better images without incurring obvious mode collapse. Furthermore, as shown in our visualization of latent code space in supervised and unsupervised tasks (see Figs 9 and 12), the code generator does exert a regularization effect while producing better images.
>
> "- I have some doubts over why AAE works so poorly when the latent dimension is 2000. How to make sure it's not a problem of implementation or the model wasn't trapped into a bad local optima / saddle points. Could you justify this?"
>
> Thanks for pointing out this. We have implemented a pair of more capable encoder and decoder with ResNets. AAE now performs reasonably well (see Figs. 5 and 6). But, still when the latent dimension is increased to 100-D or 2000-D, the simple Gaussian prior may overly regularize the model. Imagine that the latent codes generated by the encoder may occupy only a tiny portion of the high dimensional code space specified by the prior. In this case, the limited training data can hardly ensure that every random sample drawn from the prior would produce a good decoded image.

---

> > ### Author Response · Authors · 2018-01-08
> > **Response to Reviewer 3**
> >
> > Dear Reviewer 3,
> >
> > "- Contributions; this paper propose an improvement over a existing model. However, neither the idea/insights it brought can be applied onto other generative models, nor the improvement bring a significant improvement over the-state-of-the-arts. I am wondering what the community will learn from this paper, or what the author would like to claim as significant contributions. "
> >
> > Thanks for your comments.
> >
> > With the changes we have made so far, we believe our contributions include
> > 1)	We replace the simple prior with a learned prior by training the code generator to output latent variables that will minimize an adversarial loss in data space.
> > 2)	We employ a learned similarity metric (Larsen et al., 2015) in place of the default squared error in data space for training the autoencoder.
> > 3)	We maximize the mutual information between part of the code generator input and the decoder output for supervised and unsupervised training using a variational technique introduced in InfoGAN (Chen et al., 2016).
> >
> > Extensive experiments confirm its effectiveness of generating better quality images and learning better disentangled representations than AAE in both supervised and unsupervised settings, particularly on complicated datasets. In addition, to the best of our knowledge, this is one of the first few works that attempt to introduce a learned prior for AAE.
> >
> > We have re-written extensively the entire manuscript, presenting more experimental results and analyses as requested.

---

### Official Review · AnonReviewer2 · 2017-11-28
**A simple idea to improve adversarial autoencoders by learning priors**

**Rating:** 6
**Confidence:** 4

**Review:**

Recently some interesting work on a role of prior in deep generative models has been presented. The choice of prior may have an impact on the expressiveness of the model [Hoffman and Johnson, 2016]. A few existing work presents methods for learning priors from data for variational autoencoders [Goyal et al., 2017][Tomczak and Welling, 2017].  The work, "VAE with a VampPrior," [Tomczak and Welling, 2017] is missing in references.

The current work focuses on adversarial autoencoder (AAE) and introduces a code generator network to transform a simple prior into one that together with the generator can better fit the data distribution. Adversarial loss is used to train the code generator network, allowing the output of the network could be any distribution. I think the method is quite simple but interesting approach to improve AAEs without hurting the reconstruction. The paper is well written and is easy to read. The method is well described. However, what is missing in this paper is an analysis of learned priors, which help us to better understand its behavior.

The model is evaluated qualitatively only. What about quantitative evaluation?

---

> ### Author Response · Authors · 2018-01-06
> **Response to Reviewer 2**
>
> Dear Reviewer2,
>
> "Recently some interesting work on a role of prior in deep generative models has been presented. The choice of prior may have an impact on the expressiveness of the model [Hoffman and Johnson, 2016]. A few existing work presents methods for learning priors from data for variational autoencoders [Goyal et al., 2017][Tomczak and Welling, 2017]. The work, "VAE with a VampPrior," [Tomczak and Welling, 2017] is missing in references. "
>
> Thanks for your suggestion. We have cited this work in Introduction and provided a description in Related Work.
>
> "The current work focuses on adversarial autoencoder (AAE) and introduces a code generator network to transform a simple prior into one that together with the generator can better fit the data distribution. Adversarial loss is used to train the code generator network, allowing the output of the network could be any distribution. I think the method is quite simple but interesting approach to improve AAEs without hurting the reconstruction. The paper is well written and is easy to read. The method is well described. However, what is missing in this paper is an analysis of learned priors, which help us to better understand its behavior. The model is evaluated qualitatively only. What about quantitative evaluation? "
>
> Thanks for your suggestion. For quantitative evaluation, we have compared the inception score of the proposed method with other generative models in Table I. We also have visualized the learned priors with t-SNE in Figs. 9 and 12 for the supervised and unsupervised learning tasks. The text in Section 4.2.1 and Section 4.2.2 have been modified accordingly to include the discussions (see the last paragraphs in these sections).
>
> In addition, since receiving the review comments, we have improved our model in several significant ways, including
> 1) Introducing a pair of more capable encoder and decoder with ResNets. (See appendix for the implementation details)
> 2) Employing a learned similarity metric in place of the default squared error in data space to improve the convergence of the decoder. (See Section 3 Learning The Prior for the reasons)
> 3) Introducing the variational technique in InfoGAN for training the decoder and code generator when it is necessary to generate images conditionally on an input variable, as in our supervised and unsupervised learning tasks. (See Section 3 Learning The Prior for the reasons) With these changes, our model can now produce much better images without incurring obvious mode collapse.
>
> We have re-written extensively the entire manuscript, presenting more experimental results and analyses as requested.

---

### Public Comment · ~Thanh_Tung_Hoang1 · 2017-11-02
**Wasserstein GAN could improve the mode collapse problem**

AAE with code generator can produce much better images but suffer from mode collapse. It seems that the improvement in the image quality is due to the fact that the network has remembered some of the input. In other words, the mode collapse problem makes generated images look better. I would love to see the result without mode collapse problem. For example, you could try Wasserstein GAN which suffer less from mode collapse problem. I am also interested in the learned prior distribution. If you could provide some analysis on the learned prior then your paper could be much better.

---

> ### Author Response · Authors · 2018-01-06
> **Changes we've done in the revised manuscript**
>
> Dear Thanh Tung Hoang,
>
> "AAE with code generator can produce much better images but suffer from mode collapse. It seems that the improvement in the image quality is due to the fact that the network has remembered some of the input. In other words, the mode collapse problem makes generated images look better. I would love to see the result without mode collapse problem. For example, you could try Wasserstein GAN which suffer less from mode collapse problem. I am also interested in the learned prior distribution. If you could provide some analysis on the learned prior then your paper could be much better."
>
> Since receiving the review comments, we have improved our model in several significant ways, including
> 1)	Introducing a pair of more capable encoder and decoder with ResNets. (See appendix for the implementation details)
> 2)	Employing a learned similarity metric in place of the default squared error in data space to improve the convergence of the decoder. (See Section 3 Learning Priors for the reasons)
> 3)	Introducing the variational technique in InfoGAN for training the decoder and code generator when it is necessary to generate images conditionally on an input variable, as in our supervised and unsupervised learning tasks. (See Section 3 Learning Priors for the reasons)
>
> With these changes, our model can now produce much better images without incurring obvious mode collapse.
>
> We have re-written extensively the entire manuscript, presenting more experimental results and analyses.

---

### Decision · Program_Chairs · 2018-01-29
**ICLR 2018 Conference Acceptance Decision**

**Decision:**

Reject

**Comment:**

The paper proposes learning the prior for AAEs by training a code-generator that is seeded by the standard Gaussian distribution and whose output is taken as the prior. The code generator is trained by minimizing the GAN loss b/w the distribution coming out of the decoder and the real image distribution. The paper also modifies the AAE by replacing the L2 loss in pixel domain with "learned similarity metric" loss inspired by the earlier work (Larsen et al., 2015).

The contribution of the paper is specific to AAE which makes the scope narrow. Even there, the benefits of learning the prior using the proposed method are not clear. Experiments make two claims: (i) improved image generation over AAE, (ii) improved "disentanglement".

Towards (i), the paper compares images generated by AAE with those generated by their model. However, it is not clear if the improved generation quality is due to the use of decoder loss on the learned similarity metric (Larsen at al, 2015), or due to the use of GAN loss in the image space (ie, just having GAN loss over decoder's output w/o having a code generator), or due to learning the prior which is the main contribution of the paper. This has also been hinted at by AnonReviewer1. Hence, it's not clear if the sharper generated images are really due to the learned prior.

Towards (ii), the paper uses InfoGAN inspired objective to generate class conditional images. It shows the class-conditional generated images for AAE and the proposed method. Here AAE is also trained on "learned similarity metric" and augmented with similar InfoGAN type objective so the only difference is in the prior. Authors say the performance of both models is similar on MNIST and SVHN but on CIFAR their model with "learned prior" generates images that match the conditioned-upon labels better. However this claim is also subjective/qualitative and even if true, it is not clear if this is due to learned prior or due to the extra GAN discriminator loss in the image-space -- in other words, how do the results look for AAE + a discriminator in the image space, just like in the proposed model but without a code generator?

The t-SNE plots for the learned prior are also shown but they are only shown when InfoGAN loss is added. The same plots are not shown for AAE with added InfoGAN loss so it is difficult to know the benefits of learning the code-generator as proposed.

Overall, I feel the scope of the paper is narrow and the benefits of learning the prior using the method proposed in the paper are not clearly established by the reported experiments. I am hesitant to recommend acceptance to the main conference in its current form.